Exposure-related, global alterations in innate and adaptive immunity; a consideration for re-use of non-human primates in research

Bates François A. 1
Duncan Elizabeth H. 2
Simmons Monika 3
Robinson Tanisha 2 4
Samineni Sridhar 1 5
Strbo Natasa 6
Villasante Eileen 7
Bergmann-Leitner Elke 2
Wijayalath Wathsala wathsala.k.wijayalatharachchige.ctr@mail.mil 7 8
1 Veterinary Services Program, Walter Reed Army Institute of Research , Silver Spring , MD , United States of America
2 Immunology Core/Malaria Biologics Branch, Walter Reed Army Institute of Research , Silver Spring , MD , United States of America
3 Viral and Rickettsial Diseases Department, Naval Medical Research Center , Silver Spring , MD , United States of America
4 Henry M. Jackson Foundation for the Advancement of Military Medicine, Inc. (HJF) , Bethesda , MD , United States of America
5 SoBran, Inc , Falls Church , VA , United States of America
6 Department of Microbiology and Immunology, Miller School of Medicine University of Miami , Miami , FL , United States of America
7 Malaria Department, Naval Medical Research Center , Silver Spring , MD , United States of America
8 CAMRIS International , Bethesda , MD , United States of America
Villarino Alejandro
Electronic publication date: 2021 Mar 8
Publication date: 2021
Volume: 9
Electronic Location ID: e10955
Received 2020 Nov 26; Accepted 2021 Jan 27
Copyright year: 2021
Copyright holder: Bates et al.
License: This is an open access article, free of all copyright, made available under the Creative Commons Public Domain Dedication. This work may be freely reproduced, distributed, transmitted, modified, built upon, or otherwise used by anyone for any lawful purpose.
License URL: https://creativecommons.org/publicdomain/zero/1.0/

Keywords: Innate and adaptive immunity, Re-use of NHPs in research, Infectious diseases, Cytokines, Trained immunity, Non-human primates (NHPs), Global immune profile

Funding: Peer Reviewed Medical Research Program (PRMRP) Award #W81XWH-13-2-0099 Military Infectious Diseases Research Program (MIDRP) Proposals #F0580_19_WR_CS F0641_20_WR_CS Naval Medical Research Center—Silver Spring, MD N6264518C4003 This work was supported by Peer Reviewed Medical Research Program (PRMRP) Award #W81XWH-13-2-0099 (NMRC PI = Eileen Villasante) and the Military Infectious Diseases Research Program (MIDRP) Proposals # F0580_19_WR_CS and F0641_20_WR_CS (PI Elke Bergmann-Leitner). Also, this material is based upon work supported by the Naval Medical Research Center - Silver Spring, MD, under contract N6264518C4003. The funders had no role in study design, data collection and analysis, decision to publish, or preparation of the manuscript.

==============================
Background

Non-human primates (NHPs) play an important role in biomedical research, where they are often being re-used in multiple research studies over the course of their life-time. Researchers employ various study-specific screening criteria to reduce potential variables associated with subsequent re-use of NHPs. However, criteria set for NHP re-assignments largely neglect the impact of previous exposures on overall biology. Since the immune system is a key determinant of overall biological outcome, an altered biological state could be predicted by monitoring global changes in the immune profile. We postulate that every different exposure or a condition can generate a unique global immune profile in NHPs.

Methods

Changes in the global immune profile were evaluated in three different groups of rhesus macaques previously enrolled in dengue or malaria vaccine studies over six months after their last exposure. Naïve animals served as the baseline. Fresh blood samples were stained with various immune cell surface markers and analyzed by multi-color flow-cytometry to study immune cell dynamics in the peripheral blood. Serum cytokine profile in the pre-exposed animals were analyzed by mesoscale assay using a customized U-PLEX NHP biomarker panel of 12 cytokines/chemokines.

Results

Pre-exposed macaques showed altered dynamics in circulating cytokines and certain innate and adaptive immune cell subsets such as monocytes, HLA-DR+NKT cells, B cells and T cells. Some of these changes were transient, while some lasted for more than six months. Each group seemed to develop a global immune profile unique to their particular exposure.

Conclusion

Our data strongly suggest that re-used NHPs should be evaluated for long-term, overall immunological changes and randomly assigned to new studies to avoid study bias.

Introduction

Non-human primates (NHPs) are a long-lived, sentient species which can be trained to perform certain cognitive and behavioral tasks (Scott et al., 2003; Prescott, Bowell & Buchanan-Smith, 2004; Calapai et al., 2017). Most of the experiments involving NHPs are not terminal, therefore, they remain available in research facilities after the study end-points. For these reasons, it is a common practice to re-use NHPs in multiple research studies over the course of their lifetime (Tardif et al., 2013; Epstein & Vermeire, 2017). A survey conducted in North America from 2010 to 2012, showed that of the 26 facilities that responded to the survey, five facilities re-used 80% or more NHPs and seven facilities re-used 50% or more NHPs within their own facility (Lankau et al., 2014). An earlier analysis of 2,937 peer-reviewed articles published in 2001 had determined that 14.4% of the studies re-used or would re-use NHPs in various experiments (Carlsson et al., 2004). Their limited availability, ethical constraints, high cost of acquisition and maintenance can further influence repeated use of NHPs in research.

When re-assigning NHPs in various research protocols, each study typically implements their own study-specific, screening criteria to exclude previously exposed animals to a similar disease or similar environmental stimulus. As a standard practice, in most cases, NHPs are re-assigned to new studies after a short resting period. There have been no reports clearly defining the scientific basis of length of the resting period between repeated exposures of NHPs to different study protocols. Majority of NHP re-assignments assume that effect of pre-exposure can only be specific to a particular study or a disease, mostly neglecting their impact on overall biology of the animals. Since the immune system is a key determinant of overall biological outcome, changes in the global immune profile could represent an altered biological state of the animal. Here, we propose that global changes in immune system thus, could be a useful inclusion/exclusion criteria to re-use NHPs in different protocols or to determine their resting period.

The immune system continuously cross-talks with the rest of the organs in the body, modulating the immune profile accordingly, not only during diseases directly affecting the immune system, but in diseases related to other systems such as neurological or reproductive system (Bell et al., 2011; Blair et al., 2016; Raper et al., 2016). Likewise, the critical role of non-specific immunomodulatory effects induced by vaccines targeting infectious diseases have been increasingly recognized over the recent years (Kandasamy et al., 2016; De Bree et al., 2018; Blok et al., 2020). It has been shown that exposure to vaccines such as Bacille Calmette-Guerin (BCG) and small pox can induce long-lasting, heterologous immune effects mediated by epigenetic modifications in innate immune cells (trained immunity) (Benn et al., 2013; Blok et al., 2015; Arts et al., 2016; Netea et al., 2020). Acute viral or bacterial infections also seem to induce global changes in immune responses over a considerable period of time (Silveira et al., 2017; Raijmakers et al., 2019). Rhesus macaques experimentally infected with Zika virus have been shown to maintain altered frequencies of circulating monocyte and myeloid dendritic cell subsets up to seventy days post-infection (Silveira et al., 2017). During a six month follow-up, patients with acute Q fever have demonstrated long-lasting transcriptional changes in circulating monocytes and certain cytokines (Raijmakers et al., 2019). In fact, various factors such as transportation, relocation, acclimation, social conditions and aging also can adversely impact the immune profile of macaques (Didier et al., 2012; Capitanio & Cole, 2015; Nehete et al., 2017). These evidence lead to our hypothesis that every different exposure or a condition can generate a unique global immune profile in NHPs.

In order to understand whether each exposure could have resulted in a distinctive overall immune profile, we studied three different groups of rhesus macaques previously enrolled in infectious disease research protocols. Pre-exposed animals were monitored over 6 months after their most recent exposure, whereas naïve macaques served as the baseline. To avoid any technical bias, flow cytometric monitoring of immune cell frequencies and automated leukocyte counting were performed using fresh whole blood. We found long-term, exposure-related changes in circulating immune cell subsets and cytokines, leading to distinctive overall immune profiles in pre-exposed rhesus macaques.

Materials and Methods

Animals

Blood and serum samples reported in the present study were received as part of a tissue-sharing program from the protocols reviewed and approved by Walter Reed Army Institute of Research (WRAIR)/Naval Medical Research Center (NMRC) Institutional Animal Care and Use Committee in compliance with all applicable federal regulations governing the protection of animals and research (WRAIR/NMRC IACUC protocols:18-IDD-01L, 18-VD-27L, 18-VET-25L. For samples shared through WRAIR clinical pathology laboratory, 18-VET-24L). Experiments reported herein were carried out in compliance with the Animal Welfare Act and per the principles set forth in the “Guide for Care and Use of Laboratory Animals”, Institute of Laboratory Animals Resources, National Research Council, National Academy Press, 2011, the Public Health Service Animal Welfare Policy, and the policies of WRAIR. Adult purpose-bred rhesus macaques of Indian origin were housed at the WRAIR animal facility for the duration of their studies. Animals were socially pair-housed and fed a commercial diet (Lab diet 5038, Purina Mills International), provided free access to water, and supplemented with a variety of fresh fruits and vegetables. Environmental enrichment was provided in accordance with standard operating procedures of the WRAIR animal facility. Animal cages were cleaned daily and sanitized bimonthly. Automatic lighting was provided through a 12:12 h cycle.

Experimental and control groups

Three groups of rhesus macaques (Macaca mulatta, three to six years old) were selected based on their most recent study exposures. Since all the assessments were made on fresh whole blood samples, age matched, naïve rhesus macaques (n = 9, Females (F) = 3, Males (M) = 6) served as the control group for all the comparisons described in the present study. Experimental group one consisted of rhesus macaques (n = 10, F = 2, M = 8) that had been primed with bivalent or tetravalent dengue purified inactivated vaccine conjugated with alum adjuvant (DPIV, intramuscular, Day 0) and boosted with tetravalent dengue live attenuated virus (TDENV-LAV, subcutaneous, Day 28) followed by a challenge with dengue virus-2 (DENV-2, subcutaneous, Day 63). Day of the DENV-2 challenge was considered as the day of the last exposure for group one. Experimental groups two and three consisted of rhesus macaques that received a cell based, 293-gp96-Ig-PfCSP-PfAMA (gp96-Ig-PfCA) (n = 5, F = 3, M = 2) and NMRC-M3V-D/Ad-PfCSP-PfAMA (D/Ad-PfCA) (n = 5, F = 2, M = 3) candidate malaria vaccines, respectively. The gp96-Ig-PfCA malaria vaccine had contained irradiated human embryonic kidney (HEK) 293 cells transfected with a mixture of plasmids encoding a secreted form of heat shock protein gp96 (gp96-Ig), Plasmodium falciparum circumsporozoite (PfCSP) protein and P. falciparum apical membrane antigen (PfAMA). The vaccine had been administered subcutaneously in three doses at 0, 5 and 25 weeks. In D/Ad-PfCA heterologous prime-boost vaccination, macaques in the group three had been primed with three intramuscular doses containing a mixture of two DNA plasmids encoding PfCSP and PfAMA at 0, 4, 8 weeks and boosted once intramuscularly with a mixture of two non-replicating recombinant human serotype 5 adenovirus vectors expressing PfCSP and PfAMA antigens at 25 weeks. The day of last exposure for group two and three were therefore, 25 weeks post-last vaccination/boost.

Sample collection

Whole blood was collected from the femoral vein directly into EDTA collection tubes for flow cytometry analysis or into serum separation tubes. Additional serum samples were obtained through tissue sharing agreements with the relevant dengue and malaria vaccine protocols. Whole blood samples (100 µl) were processed within four hours of collection and stained cells were immediately acquired by multi-parametric flow cytometry for immunophenotyping. Serum samples (75 µl) were sub aliquoted into polypropylene micro tubes and stored at −80 °C, and later used to determine the concentration of circulating cytokines and antigen-specific antibodies. Time points were determined based on availability of samples within each experimental group.

Flow cytometry

Multi-parametric, flow cytometry was used to determine the phenotype and frequency of innate and adaptive immune cell subsets in the peripheral blood. Cell frequencies for experimental group one were measured at 1,2,3,4 and 6 months post-DENV challenge. Cell frequencies for experimental group two and three were measured at day 6, day 20, 2.5, 4, and 6 months post-last vaccination/boost. The nine color NHP immunophenotyping panel consisted of following surface antibodies from BD biosciences, San Jose, CA: CD45 (V450, clone D058-1283), CD3 (APC-Cy™7, clone SP34-2), CD8 (APC, clone RPA-T8), CD16 (PE, clone 3G8), HLA-DR (BV605, clone G46-6); following antibodies from Miltenyi Biotec, San Diego, CA; CD20 (PE-vio770, clone LT20), CD4 (Percp-Vio700, clone M-T466), CD159a (PE, clone REA 110), CD14 (FITC, clone TÜK4); and CD11c (Alexa Fluor 700, clone 3.9, Thermofisher scientific, Waltham, MA). One hundred microliters of whole blood sample was stained with a cocktail of fluorochrome-conjugated antibodies (seven color panel for group one and nine color panel for group two/three), incubated for 10 min at room temperature and diluted in sheath fluid for immediate acquisition. Acquisition was limited to cells expressing V450 fluorochrome/CD45 (trigger) at a particle cut-off size (FSC) of 4000 and 50,000 events/sample were acquired at a medium flow rate by 17-color, LSRII Fortessa flow cytometer at the ImmunoCore facility using the FACS DIVA software. Gating strategy used to analyze the phenotype and frequency of each immune cell subset is provided in the Fig. S1. Flow data was analyzed by FCS express 6.0 software.

Absolute cell counts

Concentration of white blood cells in EDTA-preserved whole blood samples was determined using a Luna Dual Fluorescence Cell Counter within an hour of sample collection (Logos Biosystems, Annandale, VA). Ten microliters of whole blood was diluted in 990 µL of 1x PBS (1:100), mixed by vortexing and 2 µL of acridine orange/propodium iodide dye mix (Logos Biosystems, Annandale, VA) was added to 18 µL of diluted blood sample. Ten microliters from the mixture was loaded into Luna photon-slide disposable hemocytometer (VitaScientific, Beltsville, MD), and the concentration of fluorescent white blood cells was determined using the Luna dual fluorescence automated cell counter. Frequency of each individual cell subset determined by flow cytometry was used to calculate their absolute counts (cells/µL) as a proportion of the total white blood cells (CD45hi cells) (or thereby as a frequency of the parent cell population). The parent population of each immune cell subset described in the manuscript is shown in Fig. S1.

Cytokine/chemokine analysis by mesoscale assay

A customized U-PLEX NHP biomarker panel of 12 cytokines/chemokines (IFN-γ, IL-1β, IL-2, IL-4, IL-5, IL-6, IL-8, IL-10, IL-12/IL23p40, MCP-1, MIP-1α, TNF-α) (Meso Scale discovery, Gaithersburg, MD) was used to analyze serum cytokine/chemokine levels per manufacturer’s instruction. All cytokines for experimental group one were measured at day 7, 2 months, and 6 months, post-DENV-2 challenge. Cytokines for experimental group two and three were measured at day 6, day 20, 2.5 months, and 6 months post-last vaccination/boost.

Enzyme-linked immunosorbent assay (ELISA) for DENV antibodies

ELISA assay was performed as described previously (Simmons, Sun & Putnak, 2016) without any modifications.

Statistical analysis

Data from the naïve group were collected independently from the three experimental groups (also, data from the two malaria groups were collected independently from the DENV group). Therefore, series of unpaired T-tests (two-tailed) were carried out to compare between the naïve group and each of the experimental groups at different time points post-exposure (alpha level of 0.05). Welch’s correction was applied with unpaired T-test, when p-value of the F test to compare variances were ≤0.05. Data approximately conformed Shapiro–Wilk test and Kolmogorov–Smirnov tests for normality at 0.05 alpha level. Data were presented as mean ± standard deviation in the text and in the figures. All statistical analysis were conducted using Graph Pad Prism 8 software. R studio (Version 1.2.1335) was used to generate correlograms (pearson correlation, two-tailed, 0.05 alpha level) and dendograms. Agglomerative hierarchical clustering with Pvclust was used to plot dendograms. AU p-values were reported on the dendograms.

Results

Innate immune cell dynamics in peripheral blood

To study innate immune cell dynamics, we monitored frequencies and absolute cell counts of circulating monocytes (CD45+CD14+ or CD45+CD14+HLA-DR+), natural killer (NK) cells (CD45+CD3-CD16/CD159a+), NK cells expressing HLA-DR (CD45+CD3-CD16/CD159a+HLA-DR+), natural killer T(NKT) cells (CD45+CD3+CD16/CD159a+), NKT cells expressing HLA-DR (CD45+CD3+CD16/CD159a+HLA-DR+) and dendritic cells (DC)s (CD45+CD14-CD16/CD159a-CD3-CD20-CD11c+HLA-DR+) in whole blood by flow cytometry and automated cell counting (Figs. S1A, S1G and S1H). DC gating strategy is not shown in the figure, but described in the legend.

Experimental group one consisted of rhesus macaques (n = 10) that had been primed with bivalent or tetravalent dengue purified inactivated vaccine (DPIV) and boosted with tetravalent dengue live attenuated virus (TDENV-LAV) followed by a challenge with dengue virus-2 (DENV-2). Experimental group two had received three doses of a cell based malaria vaccine, containing irradiated HEK 293 cells secreting heat shock protein chaperon, gp96-Ig, and the two malaria antigens, circumsporozoite protein and apical membrane antigen-1 (gp96-Ig-PfCA) (n = 5). Experimental group three had been primed thrice with a mixture of DNA plasmids expressing the same two malaria antigens as in the experimental group two, and had been boosted once with non-replicating recombinant human serotype 5 adenovirus (Ad5) plasmids expressing the same antigens (D/Ad-PfCA) (n = 5). The pre-exposed animals were monitored at multiple time points, over 6 months since their most recent exposure (ex: last dose of vaccine or challenge). A group of naïve monkeys (n = 9) (age and sex matched, see materials and methods) were used as the controls.

All the NHP monocyte subsets explicitly express CD14 (Ziegler-Heitbrock, 2014), therefore, we used the same surface marker to identify circulating monocytes in rhesus macaques in the dengue vaccinated group (Fig. S1G). Since HLA-DR has been shown to reliably identify monocytes along with CD14 (Abeles et al., 2012), we later distinguished monocytes based on CD14 and HLA-DR co-expression for the two malaria vaccine groups (Fig. S1G). Due to some previous reports on lack of expression of CD56 on rhesus macaque NK cells (Carter et al., 1999; Shields et al., 2006), here we used CD159a, a previously validated marker for rhesus NK cells (Choi et al., 2008), in combination with CD16 (Fig. S1H).

Experimental group one, which received prime-boost, DPIV/TDENV-LAV vaccine, showed elevated frequency of CD14+ monocytes at 1 month (8.3 ± 3.8, P = 0.007) and 3 months (7.6 ± 4.2, P = 0.03) post DENV-2 virus challenge compared to the naïve animals (3.9 ± 1.8). This transient expansion of monocytes were subsided to the levels present in the naïve macaques by 6 months post-challenge (Fig. 1A). While the absolute cell count data for this group is not available for early time points, we did not see any differences in the counts at late time points (4 and 6 months post-challenge) between vaccinated and naïve animals (data not shown). These data suggest that intermittent and transient alterations to circulating monocytes can be sustained up to 3 months post-DENV challenge. In contrast, no changes in the frequency and absolute cells counts of NK and NKT cell populations were observed (data not shown). For the group 1 macaques, the panel did not include surface markers for HLA-DR and DCs.

Figure 1 Monocyte dynamics in peripheral blood.

(A) CD45+CD14+ monocyte frequencies in naïve animals (n = 9) and rhesus macaques received DPIV/TDENV-LAV vaccination/DENV-2 challenge (n = 10) (B) Frequency and (C) cell counts of HLA-DR+CD14+ monocytes in naïve animals (n = 9) and the D/Ad-PfCA vaccinated animals (n = 5). Scatter dot plots show values of individual NHPs with mean ± standard deviation. Asterisks (*) denote significant differences between naïve and the experimental groups at 0.05 alpha level.

Dynamics of HLA-DR+CD14+ monocytes (both frequencies and cell counts) were found to be similar between rhesus macaques vaccinated with gp96-Ig-PfCA and naïve animals over the course of 6 months (Table S1). In contrast, animals vaccinated with D/Ad-PfCA showed a transient increase in monocyte frequencies at day 20 and at 4 months post-last boost compared to the naïve macaques (Fig. 1B, Table S1). We also noted that monocyte frequency of some of the individual macaques tend to stay consistently higher up to 4 months after their last exposure to Ad5 vectored malaria vaccine boost. Changes in circulating monocyte frequency did not impact the monocyte counts in D/Ad-PfCA exposed animals (Fig. 1C).

We did not observe any noticeable changes in NK cells, HLA-DR expressing NK cells, DC or NKT cells in the two malaria vaccine groups (data not shown). However, we noted more permanent alterations to a specific subset of NKT cells expressing HLA-DR activation marker (Fig. 2 and Table S1). NKT cells are known to share receptors expressed by both T and NK cells (Lanier, Spits & Phillips, 1992). Previous evidence also suggests expression of HLA-DR on NKT cells under certain disease conditions or changes in cytokine milieu (Saikh, Kissner & Ulrich, 2002; Almeida et al., 2019), which may play an important role in inflammation and immune regulation (Marrero, Ware & Kumar, 2015).

Figure 2 HLA-DR+NKT cells dynamics in peripheral blood.

(A) Frequency and (B) counts of HLA-DR+NKT cells in naïve animals (n = 9) and gp96-Ig-PfCA vaccinated animals (n = 5). (C) Frequency and (D) counts of HLA-DR+NKT cells in naïve animals (n = 9) and D/Ad-PfCA vaccinated animals (n = 5). Scatter dot plots show values of individual NHPs with mean ± standard deviation. Asterisks (*) denote significant differences between naïve and the experimental groups at 0.05 alpha level.

The frequency of HLA-DR expressing NKT cells were notably lower in most of the individual macaques pre-exposed to gp96-Ig-PfCA at 2.5 months and 4 months after the last vaccination (Fig. 2A and Table S1). HLA-DR+NKT cell counts were transiently reduced at 6 days post-last vaccination and returned to levels comparable to the naïve animals by 20 days. Thereafter, we observed more persistent reduction of HLA-DR+NKT cell numbers through 2.5 months, 4 months and 6 months post-exposure (Fig. 2B and Table S1).

In the D/Ad-PfCA malaria group, there was a persistent reduction of HLA-DR+NKT cell frequency through 2.5 months, 4 months and 6 months post-Ad5 boost compared to the naïve animals (Fig. 2C and Table S1). Subsequently, HLA-DR+NKT cell counts were persistently decreased through 2.5 months, 4 months and 6 months, followed by a transient decrease at 6 days post-Ad5 boost (Fig. 2D and Table S1). The data show that exposure to an adeno viral-vector based, prime-boost malaria vaccination strategy could induce more consistent and permanent alterations to this rare NKT cell subset.

In summary, dynamic changes in circulating innate immune cell subsets appeared to be unique for their past exposures (ex: vaccination strategy or pathogen). Also, the data indicate that these changes can either be transient or more permanent, which could last up to 6 months.

Circulating cytokines

To determine changes in circulating cytokine milieu at multiple time points over 6 months post-last exposure, we used a customized U-PLEX NHP biomarker panel of 12 cytokines (IFN-γ, IL-1β, IL-2, IL-4, IL-5, IL-6, IL-8, IL-10, IL-12/IL-23p40, MCP-1, MIP-1α, TNF-α). Most of the cytokines were either non-detectable or present at levels less than 2 pg/mL, except MCP-1, MIP-1α and IL-12/IL-23p40.

Monocyte chemoattractant protein-1/CCL2 (MCP-1) and macrophage inflammatory proteinα/CCL3 (MIP1-α) are two major chemokines (or a chemotactic cytokines) critical for leukocyte migration and infiltration (Deshmane et al., 2009; Bhavsar, Miller & Al-Sabbagh, 2015). MIP1-α/CCL3 plays an important role in modulating the inflammatory response, particularly in viral infections, by enhancing recruitment of leukocytes (ex. macrophages, and lymphocytes) to the site of inflammation (Bhavsar, Miller & Al-Sabbagh, 2015). We found slightly increased levels of MCP-1 in some of the individual animals exposed to DENV-2 challenge (210.3 ± 105.5, P = 0.059) by 6 months compared to naïve animals (133.9 ± 47.1) (Fig. 3A). In contrast, most of the animals had persistently decreased MIP1-α levels, 7 days through 2 months post-DENV-2 virus challenge, but failed to reach statistical significance (Fig. 3A). Both MCP-1 and MIP1-α levels in the two malaria vaccine groups were comparable to naïve animals throughout the 6 month follow-up period (Figs. 3B and 3C).

Figure 3 Circulating concentrations of MCP-1, MIP1- α and IL-12/IL-23p40 cytokines.

(A) Naïve animals (n = 9) and rhesus macaques vaccinated with DPIV/TDENV-LAV followed by DENV-2 challenge (n = 10). (B) Naïve animals (n = 9) and gp96-Ig-PfCA vaccinated animals (n = 5) (C) naïve animals (n = 9) and D/Ad-PfCA vaccinated animals (n = 5). One outlier from the naïve group was removed during IL-12/IL-23p40 analysis (n = 8). Scatter dot plots show values of individual NHPs with mean ± standard deviation. Asterisks (*) denote significant differences between naïve and the experimental groups at 0.05 alpha level.

In fact, compared to MCP-1 and MIP1-α dynamics, more pronounced persistent alterations were noted in circulating concentrations of interleukin (IL)-12/IL23-p40 cytokines in pre-exposed macaques. IL-12 and IL-23 are heterodimers that share the p40 subunit, mainly produced by dendritic cells and macrophages. IL-12 mainly regulates differentiation of T helper 1 (Th1) cells and interferon gamma (IFN-γ) cytokine production. IL-23 is crucial for maintenance and expansion of Th17 cells (Lyakh et al., 2008). IL-12/IL-23p40 cytokines were persistently lower in DPIV/TDENV-LAV vaccinated animals at 7 days (47.9 ± 14.1, P = 0.007), 2 months (59.9 ± 18.3, P = 0.01) and 6 months (65.7 ± 22.9, P = 0.03) post-DENV-2 challenge, than the naïve animals (115.7 ± 52.3) (Fig. 3A). Slower but gradual increase of mean values post-DENV-2 challenge, indicates that altered IL-12/IL-23p40 levels may be returning back to the steady state over the time (Fig. 3A). We observed a similar trend in D/Ad-PfCA group, with more transient reduction of IL-12/IL-23p40 at 6 days (59.0 ± 4.6, P = 0.01) and 20 days (65.6 ± 18.7, P = 0.03) post-Ad5 boost (Fig. 3C). This was followed by a gradual increase at 2.5 months (73.9 ± 21.2, P = 0.12), subsequently returning to the levels found in the naïve group (115.7 ± 52.3) by 6 months (120.7 ± 41.6, P = 0.8) (Fig. 3C). Rhesus macaques vaccinated with gp96-Ig-PfCA had lower mean concentrations of IL-12/IL-23p40, increasing over time post-last vaccination, yet remained statistically comparable to the naïve animals (Fig. 3B).

Dynamics of IL-12/IL-23p40 seemed to follow a trend unique to each exposure, when shifting from altered state to steady or naïve state. This transition may take up to 2.5 months or longer than 6 months post-last exposure, depending on the type of the exposure. Our findings agree with the fact that different exposures can induce either transient or persistent changes, uniquely altering the dynamics and composition of the circulating cytokine milieu.

B cells

In order to evaluate global impact on the humoral arm of the adaptive immunity, we determined the cell frequency and counts of circulating B cells. Interestingly, a persistent reduction in the overall CD20+B cell frequency was seen after DENV-2 challenge (Fig. 4A and Table S2). Although data are not available for the first three time points (1, 2 and 3 months), B cell counts at 4 months and 6 months post-challenge were still significantly lower than the naïve group (Fig. 4B and Table S2). It appears that the majority of circulating B cells may have been either suppressed or have migrated to the secondary lymphoid organs, presumably as a result of the prime-boost DPIV/TDENV-LAV vaccination. High titers of antibodies against all the four dengue serotypes (1-4) were detected at 2 months and 6 months post-DENV-2 challenge (Fig. S2A). The ability of DPIV/TDENV-LAV vaccine strategy to elicit high levels of DENV 1-4 specific antibodies may likely be associated with the dynamics of circulating B cells.

Figure 4 B cell dynamics in peripheral blood.

(A) Frequency and (B) counts of B cells in naïve animals (n = 9) and in rhesus macaques vaccinated with DPIV/TDENV-LAV followed by DENV-2 challenge (n = 10) (C) frequency and (D) counts of B cells in naïve animals (n = 9), gp96-Ig-PfCA vaccinated animals (n = 5) and D/Ad-PfCA vaccinated animals (n = 5). Data represent mean ± standard deviation. Asterisks (*) denote significant differences between naïve and the experimental groups at each time point (0.05 alpha level).

As shown in Fig. 4C, the D/Ad-PfCA group also had decreased B cell frequencies at 6 days and 20 days, which then became comparable to the naïve animals by 2.5 months post-Ad5 boost. Our data suggest that this transient stabilization after the acute phase of peripheral immune response is followed by a second wave of more persistent alterations, as the D/Ad-PfCA group showed persistently reduced B cell frequencies at 4 months and 6 months (Fig. 4C and Table S2). Variations in the B cell frequencies in D/Ad-PfCA vaccinated animals subsequently affected the B cell counts up to 4 months post-Ad5 boost. Nevertheless, by 6 months, B cell counts of D/Ad-PfCA group were comparable to the naïve animals, despite the differences observed in B cell frequencies (Fig. 4D and Table S2). In contrast to the D/Ad-PfCA group and DENV group, B cell frequencies and counts in gp96-Ig-PfCA vaccinated animals were maintained within the range found in the naïve animals (Figs. 4C, 4D and Table S2).

Based on the differential ability of DPIV/TDENV-LAV, D/Ad-PfCA and gp96-Ig-PfCA vaccination strategies to elicit antibodies (Figs. S2A and S2B), it may be reasonable to speculate that there is a long-term interplay between circulating B cells and antigen-specific antibody production. As we have seen with the other components of the immunity in the present study, changes in global peripheral B cell population also appeared to be exposure-specific.

T cells

T cells are the central players of cell mediated immunity, which can modulate the function of many immune cells, activate host defense mechanisms and lyse pathogen infected cells. To understand global changes in circulating T cells following exposure to different protocols, we determined the frequency and cell counts of various T cell subsets. CD3+ T cell subsets were analyzed on CD3+ gated population (Figs. S1C–S1F).

We found decreased CD3+ gated CD8+ T cell frequencies in DPIV/TDENV-LAV vaccinated animals, up to 1 month DENV-2 post-challenge (15.6 ± 8.7, P = 0.003), compared to the naïve group (26.6 ± 2.0) (Fig. 5A). This contraction appeared to be associated with the expansion of double negative (DN) T cells (vaccinated group-23.9 ± 6.3 p = 0.0002; Naïve–11.9 ± 4.0) (Fig. 5A). Frequencies of CD8+ T cells and DN cells were returned to the naïve levels after 1 month. Notably, frequency of CD3+ T cells and CD3+ gated CD4+ T cells remained unchanged throughout the follow-up (data not shown). Also, the T cell counts obtained for each cell subset at 4 months and 6 months for the vaccinated group did not differ from the naïve animals (data not shown) (data is not available for the first three months).

Figure 5 T cell dynamics in peripheral blood.

(A) Cell frequency of CD3+ gated CD8+ T cells (CD3+CD8+) and double negative T cells (CD3+CD8-CD4-) in naïve group (n = 9) and in DPIV/TDENV-LAV vaccinated animals at 1 month post-DENV-2 challenge (n = 10). (B) Frequency and (C) counts of CD3+ gated CD4+ T cells (CD3+CD4+) in naïve group (n = 9) and D/Ad-PfCA vaccinated animals (n = 5). Scatter dot plots show values of individual NHPs with mean ± standard deviation. Asterisks (*) denote significant differences between naïve and the experimental groups at 0.05 alpha level.

Gp96-Ig-PfCA vaccination did not induce any alterations to the circulating T cells, including CD3+ T cells, CD3+ gated CD4+ T cells and CD8+ T cells (data not shown). Figs. 5B and 5C show that both cell frequency (53.7 ± 4.1, P = 0.02) and cell counts (1753 ± 803, P = 0.04) of CD3+ gated CD4+ T cell population were transiently decreased by day 20 post-Ad5 boost, subsequently returning to the levels seen in naïve animals (Frequency 60.4 ± 5.0, counts-3808 ± 1958) by 2.5 months. Nevertheless, CD4+ T cell counts seemed to continuously increase over 6 months post-Ad5 boost (within the range seen in naive animals), rather than remaining in a steady state (Fig. 5C). CD3+ T cells and CD3+ gated CD8+ T cell populations were not affected by D/Ad-PfCA vaccination (data not shown).

We then analyzed the dynamics of CD3+ T cell subsets expressing HLA-DR, a well-known marker for activated T cells (Shipkova & Wieland, 2012). We observed a gradual reduction of frequency and cell counts of activated T cells (CD3+HLA-DR+), CD8+ T cells (CD3+ gated CD8+HLA-DR+) and CD4+ T cells (CD3+ gated CD4+HLA-DR+), over 6 months post-last gp96-Ig-PfCA vaccination compared to naïve animals (Figs. 6A, 6B and Table S3). Impact of vaccination was more profound on frequency and cell count of activated CD8+ T cells, than on the activated CD4+ T cell population, from 2.5 months through 6 months post-last boost (Figs. 6A, 6B and Table S3). Long-term alterations to activated T cell subsets, particularly the CD8+ T cells, may likely be associated with the ability of gp96-Ig based vaccines to activate cellular arm of the immunity and to induce CD8+ T cell mediated tissue-resident memory (Strbo et al., 2011; Strbo et al., 2016; Strbo et al., 2020).

Figure 6 Dynamics of HLA-DR expressing, activated T cells in peripheral blood.

(A) Frequency and (B) cell counts of activated T cell subsets expressing HLA-DR in naïve (n = 9) and gp96-Ig-PfCA vaccinated (n = 5) animals. (C) Frequency of activated CD3+T cells expressing HLA-DR (CD45+CD3+HLA-DR+) in naïve animals (n = 9) and D/Ad-PfCA vaccinated animals (n = 5). (D) Frequency of CD3+ gated, CD4+ T cells expressing HLA-DR (CD3+CD4+HLA-DR+) in naïve animals (n = 9) and D/Ad-PfCA vaccinated animals (n = 5). Bar graphs show mean and the standard deviation. Scatter dot plots show values of individual NHPs with mean ± standard deviation. Asterisks (*) denote significant differences between naïve and the experimental groups at 0.05 alpha level.

As shown in Fig. 6C, after Ad5 boost, we found elevated frequency of activated CD3+ T cells by 20 days, followed by a gradual decrease to naïve levels (Table S4). In contrast, activated CD4+ T cells (CD3+ gated) maintained higher mean frequencies than naïve animals throughout the 6-month follow-up period, where the values were statistically significant at 20 days and 2.5 months (Fig. 6D and Table S4). Changes in frequencies of activated CD3+ T cells and CD3+ gated CD4+ T cells did not alter their cell counts (Table S4). Dynamics of activated CD8+ T cells remained comparable to naïve animals (Table S4). Our data indicate that the D/Ad-PfCA malaria vaccine mainly alter the CD4+ T cell component in circulation, inducing persistent changes over 6 months post-last exposure.

Here we show that each individual exposure could alter the global balance of cell mediated adaptive immune responses in the peripheral blood, irrespective of the anticipated vaccine-focused, antigen-specific, cell-mediated immune responses.

Global immune profile

We next sought to determine whether the observed changes in individual immune cell subsets could subsequently affect the overall immune profile of pre-exposed rhesus macaques (Fig. 7). We used correlation matrices and hierarchical clustering to visualize the relationships between cell counts of different cell types within naïve (Fig. 7A), gp96-Ig-PfCA (Fig. 7B) and D/Ad-PfCA (Fig. 7C) groups. The immune system consists of a highly integrated network of immune cell subsets, where changes in cell counts of a certain immune cell subset could be positively or negatively related to the counts of another immune cell type, particularly after exposure to different antigenic stimuli. For example, counts of certain circulating immune cell subsets could be synergistically altered due to leukocyte migration during an inflammatory immune response. These associations subsequently represent the overall immune profile of an individual.

Figure 7 Correlation analysis and hierarchical clustering of immune cell subsets of naïve and pre-exposed animals.

Correlograms show the correlation between cell counts (cell/µL) of circulating immune cell subsets, within each of the following groups; (A) Naïve (n = 9) (B) gp96-Ig-PfCA vaccinated (n = 25) and (C) D/Ad-PfCA vaccinated (n = 25) animals. Data from naïve animals represent only a single time point. For the two vaccinated groups, samples were pooled from the 5 time points post-last exposure (n = 5 × 5); 6 days, 20 days, 2.5 months, 4 months and 6 months to capture the immune cell dynamics over 6 months. DENV group was not included in the correlation analysis, due to unavailability of cell count data for all the time points. Positive correlations are displayed in blue and negative correlations in red color. Color intensity and the size of the circle are proportional to the correlation coefficients (r). Dendograms display hierarchal relationship between immune cell subsets produced by correlation analysis. Agglomerative hierarchical clustering integrated with complete linkage method (distance—Euclidean, Pvclust) was used to build the immune cell clusters. Each leaf of the dendrogram corresponds to one immune cell subset. Closely associated cell subsets are fused into branches. Similarity of the association between two immune cell subsets decreases as height of the fusion increases along the vertical axis. Values at branches are AU p-values (approximately unbiased (AU) probability values (%) by multiscale bootstrap resampling). Clusters with AU > =95% are considered to be strongly supported by data and highlighted by the red rectangles. We used cell counts calculated based on the cell frequencies of helper T (CD45+CD4+) and cytotoxic T cells (CD45+CD8+) directly gated on parent CD45+ leukocyte population, instead of sub-gating on CD3+ T cell population for correlation analysis and hierarchical clustering (Fig. S1B).

Correlation coefficient of cell counts was computed for all possible combinations (pairs) of immune cell subsets within each group to generate a correlation matrix and was visualized by a correlation plot (correlogram) using R studio software (see methods-statistical analysis). Each correlogram represents negative and positive relationships between the cell counts of all pairs of immune cell subsets tested within each group (Fig. 7). Correlograms of the two vaccinated groups, gp96-Ig-PfCA (Fig. 7B) and D/Ad-PfCA (Fig. 7C) displayed a remarkable deviation from the naïve group (Fig. 7A). Dendrograms were generated using hierarchical clustering to achieve a higher resolution view of these relationships. Here, the cell clusters represent closely correlating measurements where the length of the clades represents the extent of similarity and closeness between these measurements (Fig. 7, dendograms). Both naïve animals (Fig. 7A) and the D/Ad-PfCA group (Fig. 7C) had two major immune cell clusters (AU=100, highlighted by a red rectangle). The two groups shared one of these major clusters containing NKT-NK cell clade (AU = 100). However, the three T cell clades in the second cluster showed distinctive differences between the two groups. More interestingly, the gp96-Ig-PfCA group had five distinctive clades, with only one strongly associated CD8+ T cytotoxic-NKT cells cell cluster (AU = 100, highlighted by red rectangle) (Fig. 7B). Differentially associated immune cell clusters, unique to each group, indicate a distinctive interplay among immune cell components based on their antigenic exposure. Thus, our data collectively suggest that each group could develop a unique global immune profile in peripheral blood under different experimental conditions.

Discussion

Repeated exposure to various physical and mental stimuli could modulate overall biology of NHPs, with possible implications on re-using those animals between various research protocols. However, so far, we have lacked proper scientific evidence to justify the widely accepted, arbitrarily implemented, 4–6 weeks rest-period between re-assignment of NHPs in different research protocols. One way to understand the impact on overall biology due to a certain exposure is to study dynamics of key biological systems of the exposed NHPs. Changes in the immune system in fact, are better indicators of overall biological impact, due to its continuous cross-talk with the rest of the body compartments (Zmora et al., 2017; Limanaqi et al., 2019; Poggi et al., 2019; Singbartl, Formeck & Kellum, 2019). Here, we show that exposure of rhesus macaques to various antigenic stimuli could largely alter overall dynamics of the immune system in peripheral blood. The alterations were unique to each exposure. While some of the alterations were transient, some changes lasted for more than six months, emphasizing the need for closer and case-by-case analysis of animal’s overall immunity, before re-assigning them to new protocols.

An increasing body of evidence suggests that exposure to various environmental insults could induce epigenetic modifications leading to sustainable changes in transcriptional programming of cells (Hamada et al., 2019). This in fact, is believed to alter the functional state of certain innate immune cells, such as monocytes and cytokines, which is sustained for weeks or months after eliminating the initial antigenic or microbial stimulus (Netea et al., 2016; Hamada et al., 2019; Zhang & Cao, 2019). Previous studies have demonstrated long-term functional reprogramming of myeloid cells following vaccinations and acute viral infections (Kleinnijenhuis et al., 2012; Yao et al., 2018; Aegerter et al., 2020). A similar phenomenon would presumably explain the alterations seen in the dynamics of circulating monocytes and cytokines that occurred over 6 months following DENV-2 challenge and Ad5 boost (Figs. 1 and 3). For example, DPIV/TDENV-LAV vaccination and/or DENV-2 challenge could reprogram the transcription of circulating MIP1-α and IL-12/IL-23p40 cytokines, leading to a long-term suppression of their secretion (Fig. 3A). MIP-1α has been shown to play a role in DENV immunopathology (Spain-Santana et al., 2001), therefore, observed suppression of MIP-1α, would presumably be related to preventing progression of dengue viral infection in vaccinated animals. Likewise, IL-12/IL-23p40 suppression seen following DENV-2 challenge may likely have been associated with cross-interference between RIG-I–like receptor (RLR) and Toll-like receptor (TLR) signaling pathways in innate immune cells (Negishi et al., 2012; Sprokholt, Helgers & Geijtenbeek, 2017). Although, IL-12 is crucial for Th1 differentiation to fight intracellular pathogens such as viruses (Komastu, Ireland & Reiss, 1998), only IFN-α and IL-27 appear to mediate Th1 polarization following DENV infections (Sprokholt et al., 2017a; Sprokholt et al., 2017b). Nevertheless, the functional relevance of the observed changes in terms of development of innate immunological memory leading to “trained immunity” or “tolerance” (Rodriguez, Suarez-Alvarez & Lopez-Larrea, 2019) has yet to be elucidated. Non-specific heterologous effects of some of the widely-used live attenuated vaccines such as BCG and measles are now thought to be mediated by innate immune memory (Arts et al., 2016). Similar effects would lead to false conclusions, even if the NHPs are being re-used in unrelated pre-clinical vaccine studies.

Besides all the non-specific, innate immune effects, exposure to antigens or microbes always unfolds a series of antigen-specific immunological effects shaping adaptive immunity. Indeed, as we observed, these antigen-specific effects seem to overtly dominate the steady state of B cell dynamics, lasting for months, more profoundly after DPIV/TDENV-LAV-DENV-2 and to some extent after DNA/Ad-PfCA exposures (Fig. 4). We also believe that persistent reductions observed in B cells more likely have been associated with maintenance of antigen-specific antibodies in the two vaccinated groups. In fact, short-lived, anti-PfCSP antibodies were produced in smaller quantities relative to more persistent, larger quantities of anti-DENV antibodies (Fig. S2), which might explain the more profound decrease of B cells in the dengue vaccinated group (Figs. 4A and 4B). Theoretically, putative antigen depots displayed in follicular dendritic cells could allure naïve B cells from the circulation contributing to the observed dynamics (Heesters et al., 2016; Kranich & Krautler, 2016). It is also plausible that memory B cells from the circulating pool may continuously be recruited to replenish the pool of plasma cells (Traggiai, Puzone & Lanzavecchia, 2003). While the exact reasons for these altered B cell dynamics are unknown, we believe that as with B cells, alterations seen in the T cell dynamics may have been also pre-programmed by the particular vaccination strategy.

More intriguingly, the fact that the different antigenic stimuli could generate unique patterns of associations between various immune cell subsets (Fig. 7) further validates our hypothesis of exposure-specific, global immune imprinting in rhesus macaques. In this modified immunological state, repeated use of pre-exposed animals may affect consistency and reproducibility of subsequent research data. Instead, our findings necessitate the development of more detailed screening criteria for re-used NHPs prior to enrollment in any subsequent protocol. The 9-color flow cytometry panel integrated with 12 U-Plex NHP cytokine panel used in this study provides a basic tool for immune-profiling. Antibody panel could be customized to detect additional immune cell subsets such as memory and regulatory cells based on the experimental needs or user preference. We recommend analysis of fresh whole blood samples, instead of purified peripheral blood mononuclear cells (PBMC), to avoid technical bias generated by purification, freeze-thawing of cells, etc. Our data warrant regular collection of blood and serum samples at baseline and at several time-points post-last exposure to evaluate, long-term dynamics of immune cells and cytokines, particularly after major immunological stressors, whether they be resultant from experimentation, social dynamics, or transportation stress. More specifically, our findings lay the groundwork for further investigations when considering re-use of pre-exposed rhesus macaques in infectious disease research. However, the present study was limited to groups of rhesus macaques housed at the WRAIR animal facility, which have been exposed to three experimental vaccines, and only one group was challenged with a live pathogen. Further research is required to determine the repeatability of these findings, particularly after exposure to other infectious agents, vaccinations, and novel therapeutics. Exposure to live pathogens would undoubtedly lead to more robust immune responses, which may result in immediate and prolonged changes in the immune cell dynamics (Silveira et al., 2017). Immunological changes observed in pre-exposed NHPs may also vary by the animal facility, NHP colonies, and the NHP species. Each animal facility has different environmental stressors/stimuli, where certain containments or facilities are maintained under germ-free conditions. Genetic variations in major histocompatibility complex exist among different NHP species or sub-populations (Viray, Rolfs & Smith, 2001; Heijmans, De Groot & Bontrop, 2020). This would potentially generate species or population-specific immune responses even when exposed to the same antigenic/environmental stimuli. Therefore, aforementioned variables should be carefully considered when developing flow-cytometry antibody panels or cytokine panels to screen re-used NHPs. The strategy of immune-profiling NHPs prior to incorporating them into experimental studies would have two interrelated benefits –more consistent experimental results by randomly re-assigning pre-exposed animals into new studies to avoid study bias and improved adherence to the 3Rs of research. Use of NHPs with less immunological variations, thereby presumably less overall biological variations, would provide better pre-clinical models in biomedical research. This, in turn, would allow the use of fewer NHPs and maximize the information obtained from them consistent with Animal Welfare Act and the reduction principle of the 3Rs.

Conclusions

Every different antigenic exposure or environmental stimulus can generate a unique global immune profile in NHPs, potentially impacting their overall biology and subsequent re-use in research experiments. To test this hypothesis we studied dynamics of circulating immune cell subsets (innate and adaptive) and cytokines in rhesus macaques previously exposed to different antigenic stimuli. We observed exposure-specific, transient and persistent changes (up to 6 months), in different immune cell populations as well as in the cytokine profile. Our data suggest that overall biological changes represented by exposure-related, global immune imprinting should be carefully evaluated before re-assigning NHPs to new studies to avoid study bias. Further research is warranted to understand whether such changes could lead to “trained-immunity” or “immune tolerance” and subsequent impact of NHP re-use on the experimental outcome.

Supplemental Information

Data S1 Raw data

This file containes raw data for Figure 1-7.

Click here for additional data file.

Table S1 Statistics of frequency and cell counts of innate immune cells

Click here for additional data file.

Table S2 Statistics of frequency and cell counts of B cells

Click here for additional data file.

Table S3 Statistics of frequency and cell counts of activated T cells in naïve group and gp96-Ig-PfCA vaccinated animals

Click here for additional data file.

Table S4 Statistics of frequency and cell counts of activated T cells in naïve group and D/Ad-PfCA vaccinated animal

Click here for additional data file.

Figure S1 Multi-color flow-cytomtery gating strategy to identify innate and adaptive immune cell subsets in peripheral blood

Acquisition was limited to cells expressing V450 fluorochrome/CD45 (common leukocyte antigen) (trigger) at a particle cut-off size (FSC) of 4000. Diluted whole blood samples were acquired at a medium flow rate (50,000 events/sample) by 17-color, LSRII Fortessa flow cytometer at the WRAIR Flow Cytometry core facility. (A) After excluding doublets, leukocytes were gated on side-scatter and CD45. Of the two CD45 populations shown in S1A here, only the CD45hi population expressed all the other lineage markers for innate and adaptive immune cells subsets. Therefore, CD45hi population was selected for further characterization of following cell subsets (B) CD45+ gated CD4+ T cells and CD8+ T cells (C) CD45+ gated CD20+ B cells and CD45+ gated CD3+T cells (D) CD3+ gated activated HLA-DR+CD3+ T cells (E) CD3+ gated CD4+ T cells, CD3+ gated CD8+ T cells, CD4-CD8- double negative (DN) T cells (F) CD3+ gated, activated HLA-DR+CD4+ T cells and HLA-DR+CD8+ T cells (G) CD45+ gated CD14+ monocytes and CD45+ gated HLA-DR+CD14+ monocytes (H) CD45+ gated CD3-CD16/CD159a+ NK cells, CD45+ gated CD3-CD16/CD159a+HLA-DR+ NK cells, CD45+ gated CD3+CD16/CD159a+ NKT cells, CD45+ gated CD3+CD16/CD159a+HLA-DR+ NKT cells. Gating strategy for dendritic cells (DCs) is not shown here. CD45+ leukocytes co-expressing HLA-DR and CD11c were considered as DCs after excluding the cells expressing CD14 and CD16/CD159a, CD3 and CD20 (CD45+CD14-CD16/CD159a-CD3-CD20-CD11c+HLA-DR+).

Click here for additional data file.

Figure S2 Serum antibody titers (IgG)

(A) Specific to each of the four dengue serotypes (DENV 1-4) in DPIV/TDENV-LAV vaccinated animals (n=10) at 2 months and 6 months post-DENV-2 challenge (B) specific to Plasmodium falciparum (Pf) full-length circumsporozoite (CSP) protein (PfCSP) in gp96-Ig-PfCA vaccinated animals (n=5) and D/Ad-PfCA vaccinated animals (n=5) at 6 days, 20 days and 2.5 months post-last vaccination. Data represent mean and the standard deviation.

Click here for additional data file.

We sincerely thank Ms Dawn Wolf, Ms Marcia Caputo and all the staff at Veterinary Services Program, Walter Reed Army Institute of Research, for their extensive support with non-human primate work reported in this article. We also thank Dr Martha Sedegah, Ms Noelle Patterson and members of the Malaria Department, Naval Medical Research Center for sharing samples, Dr Gregory Gromowski at Viral Diseases Branch, Walter Reed Army Institute of Research for sharing samples and background information on the dengue vaccination strategy reported in this article. FAB is a military service member and EHD, MS, EV and EBL are US Government employees. The views expressed in this article are those of the authors and do not necessarily reflect the official policy or position of the Department of the Navy, Department of Defense, nor the U.S. Government. Material has been reviewed by the Walter Reed Army Institute of Research. There is no objection to its presentation and/or publication. The opinions or assertions contained herein are the private views of the authors, and are not to be construed as official, or as reflecting true views of the Department of the Army or the Department of Defense. Also, the views expressed in this article do not necessarily reflect the official policy or position of The Henry M. Jackson Foundation for the Advancement of Military Medicine, Inc. (HJF). For military service members or employees of the U.S. Government this work was prepared as part of their official duties. Title 17 U.S.C. §105 provides that ‘Copyright protection under this title is not available for any work of the United States Government.’ Title 17 U.S.C. §101 defines a U.S. Government work as a work prepared by a military service member or employee of the U.S. Government as part of that person’s official duties.

Additional Information and Declarations

Competing Interests

Author Contributions

Animal Ethics

Data Availability

Natasa Strbo is a co-inventor on patents describing gp96-Ig technology and a member of Heat Biologic’s COVID-19 Advisory Board. Other authors declare no competing interests. Sridhar Samineni is employed with SoBran, Inc., Wathsala Wijayalath is a contractor for CAMRIS International and Tanisha Robinson is a contractor for Henry M. Jackson Foundation for the Advancement of Military Medicine, Inc. (HJF).

François A. Bates conceived and designed the experiments, performed the experiments, analyzed the data, prepared figures and/or tables, authored or reviewed drafts of the paper, and approved the final draft.

Elizabeth H. Duncan, Monika Simmons and Tanisha Robinson performed the experiments, authored or reviewed drafts of the paper, and approved the final draft.

Sridhar Samineni conceived and designed the experiments, authored or reviewed drafts of the paper, and approved the final draft.

Natasa Strbo and Eileen Villasante conceived and designed the experiments, authored or reviewed drafts of the paper, and approved the final draft.

Elke Bergmann-Leitner conceived and designed the experiments, analyzed the data, authored or reviewed drafts of the paper, and approved the final draft.

Wathsala Wijayalath conceived and designed the experiments, performed the experiments, analyzed the data, prepared figures and/or tables, authored or reviewed drafts of the paper, and approved the final draft.

The following information was supplied relating to ethical approvals (i.e., approving body and any reference numbers):

Blood and serum samples reported in the present study were received as part of a tissue-sharing program from the protocols reviewed and approved by Walter Reed Army Institute of Research (WRAIR)/Naval Medical Research Center (NMRC) Institutional Animal Care and Use Committee in compliance with all applicable federal regulations governing the protection of animals and research (WRAIR/NMRC IACUC protocols:18-IDD-01L, 18-VD-27L, 18-VET-25L. For samples shared through WRAIR clinical pathology laboratory, 18-VET-24L). Experiments reported herein was carried out in compliance with the Animal Welfare Act and per the principles set forth in the “Guide for Care and Use of Laboratory Animals,” Institute of Laboratory Animals Resources, National Research Council, National Academy Press, 2011, the Public Health Service Animal Welfare Policy, and the policies of WRAIR.

The following information was supplied regarding data availability:

Raw data are available in the Supplemental Files.

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
