# Peer review of "Exposure-related, global alterations in innate and adaptive immunity; a consideration for re-use of non-human primates in research"

_PeerJ, doi:10.7717/peerj.10955_

## Round 0.1 · original submission · Minor Revisions

Please note that the suggested revisions are mainly about clarifying methods and editing the manuscript. Additional experimental data would strengthen the work but are not necessary at this time.

Reviewer 1 ·

Basic reporting

See comments below

The article is well written and the introduction was clear.

Experimental design

See comments below

I highlight point 4 - I had difficulty in understanding how the data in figure 7 was generated. My best interpretation is that the authors correlated cell counts or cell percentages in the peripheral blood of three groups of animals.

It gives the impression that there is some functional or (epi)genetic link that generates the correlations. If this is not so that the data should be presented in a less misleading manner.

Validity of the findings

See comments below.

Fundemenatly the premise is interesting and that animals should be assessed prior to starting a new experiment. Experienced animals may be more useful sample as this reflects animal populations in the real world.

Additional comments

Dear Editor,

Thank you for asking me to review the manuscript “Exposure related global alterations in innate and adaptive immunity; a consideration for re-use of non-human primates in research” by Francois Bates et al.

The authors compare four groups of rhesus macaques, naïve animals, animals that were vaccinated with a live attenuated dengue virus, those that were vaccinated with a malaria antigen and those that were vaccinated with a non-replicating human adenovirus containing the same malaria antigens.
The authors tracked changes to serum cytokines, CD14+ monocytes, HLA-DR+ NKT cells, CD20+ B cells, total and HLA-DR+ Th and Tc cells over a six month period. They found residual changes to all of these measurements that lasted for up to six months post challenge.

The authors highlight an important point in non-human primate research and suggest that animals exposed to multiple experiments will diverge from naïve animals that may confound future experiments. Conversely, they may make the animals more representative of a human population that is constantly being immunologically challenged.

A few points should be considered:
(1) There are inconsistencies in the investigation of post DENV and post malaria/ adenov vaccine effects in the monocyte (figure 1) and T cell subsets. In the former CD14+ cells were compared with DR+CD14+ cells. In figure 5 CD4+ T cell data was not shown in the DENV group.
(2) It is unclear what total populations of cells the percentage cell frequency is determined throughout the paper. For example, I assume that the % frequency of total T cells is derived from a different cell total population that that used to calculate the % CD3+ CD8+ T cells.
(3) The lymphocyte analysis is relatively superficial. It would be useful to see naïve and memory B cell subsets. It would be useful to see CD25+CD4+ T cell subsets as this would have relevance in auto immune models.
(4) I am confused by the data generated in figure 7. I do not understand how correlations can be generated on the basis of cell counts and how this could be interpreted to correlate with immune cell function.

Reviewer 2 ·

Basic reporting

In this article the authors make a case that NHPs that are re-used for multiple experiments should be carefully evaluated before assigning to experimental groups in order to avoid biassing and to enhance reproducibility. In order to evaluate the impact of prior immune stimulation on the resting immune response the authors used blood drawn from three groups of rhesus macaques at defined time points after immunization and challenge with dengue virus or after immunization or boosting with two different malaria vaccine regimens. Blood samples were taken as far as 6 months after their last exposure to antigen and compared against age and sex-matched naïve controls. The authors noted transient as well as long-lasting changes in immune cell frequency, cell numbers and cytokine production in the blood. The overall changes in biology varied depending on the type of antigenic stimuli each group was exposed to. Hence, the authors conclude that antigenic stimulation of the immune system leaves unique patterns of immunological imprints in NHPs that could affect the consistency of subsequent data when the NHPs are re-used in separate experiments. The authors suggest that their method of evaluating cytokines and immune cell phenotypes in fresh blood samples derived from NHPs at various times after the conclusion of their study will help aid in determining an appropriate time when these NHPs can be re-used in new experiments.

This is a well written manuscript that provides a clear objective, results and discussion of relevant findings and interpretation of the findings. The authors do a sufficient job of using the literature to support their study design and rationale, and their conclusions. The results were adequately summarized at the end of each results section, making it easy for the reader to understand the significance of the findings.

Experimental design

The authors provide the necessary information regarding the use of the NHPs in this study. Specific animal protocol numbers for the studies that the NHPs were apart of were provided in the methods section. A statement indicating that all procedures and protocols were approved by the institutes IACUC was provided in the methods. Also, an animal ethics statement was provided in the methods section. The authors provided information about the sex, age and number of rhesus macaques used in each group in the study in the methods section. Lastly, the research question was defined and addressed adequately by the authors in the manuscript.

Validity of the findings

While the authors emphasize that epigenetic changes in innate immune cell populations such as monocytes could impact future immunological readouts in secondary experiments due to the phenomenon of trained immunity, they do not investigate this idea in this study. However, they do a sufficient job of using the literature to support this idea in the introduction and discussion sections of the manuscript. They also allude to the need of future studies to determine how different antigenic stimuli can induce tolerance or trained immunity.

One point that the authors should briefly address in their discussion is in regard to the limitations of their study. This was one study conducted using rhesus macaques from their facility at Walter Reed. Although the use of three separate groups exposed to three different antigenic stimuli increased the rigor of the study, this is just one study. Do the authors anticipate that that other NHP species will response similarly as the rhesus macaques? Only one of the three groups was challenged with the agent that the NHPs were vaccinated against. Would similar results as seen with group one be predicted after live challenge with other pathogens? Would the authors anticipate similar findings using blood derived from NHPs used in studies conducted at different animal facilities?

---

## Round 0.2 · accepted · Accept

This is important work and I'm glad to see it published in PeerJ.